# Synchronizing Probabilities in Model-Driven Lossless Compression

**Aviv Adler**
Analog Garage
Analog Devices, Inc.
Boston, MA 02110, USA
Aviv.Adler@analog.com

**Jennifer Tang**
Department of Mathematics & Computer Science
College of the Holy Cross
Worcester, MA 01610, USA
jtang@holycross.edu

## Abstract

It is well-known in the field of lossless data compression that probabilistic next-symbol prediction can be used to compress sequences of symbols. Deep neural networks are able to capture rich dependencies in data, offering a powerful means of estimating these probabilities and hence an avenue towards more effective compression algorithms. However, both compressor and decompressor must have exactly matching predictions; even small differences from non-determinism (which often happen with learned models due to hardware, software, or computation order) can lead to cascading decoding failures. In this paper, we formalize the problem of prediction mismatch in model-driven compression, and introduce Probability Matching Interval Coding (PMATIC), a model-agnostic algorithm that tolerates bounded prediction mismatch with low overhead. PMATIC works with the predicted probabilities, making it compatible as a drop-in replacement for the arithmetic encoder in model-driven compression tools. We show theoretical correctness and performance bounds for PMATIC, and validate these results on text data. These results confirm that, when paired an advanced prediction model, PMATIC is robust to prediction mismatch while achieving compression rates that out-perform standard modern compression tools.

## 1 Introduction

### 1.1 Model-Driven Lossless Compression

A key task in modern information systems is data compression, the process of reducing the size of text, images, video, or other data so it can be stored and transmitted more efficiently. In lossless compression, the data is encoded into a compact representation from which the original can be decoded exactly, in contrast to lossy compression, which only permits approximate reconstruction. Compression is generally formalized as the problem of encoding a string of discrete symbols drawn from a finite alphabet. In deep learning contexts, these symbols are often referred to as tokens. The choice of symbols is domain-dependent: for text, tokens are typically subword units or characters; for images, they may correspond to pixel intensities, color values, or transformed coefficients; and for other domains, analogous discrete representations are used.

Lossless compression works by exploiting regularities in the data: common patterns are assigned shorter codes, while rare patterns receive longer ones. These regularities may reflect simple statistics, such as symbol frequencies, or more complex and context-dependent structure and even semantic information. From this perspective, any lossless compression method implicitly defines a probabilistic model of the data source, with compression effectiveness depending on how well the model matches the true distribution. Some algorithms make this explicit, using predictive models that estimate the probability of each symbol given its context [Cleary & Witten (1984)] in order to generate the code; we refer to such algorithms as *model-driven*. Others, such as Lempel–Ziv–Welch (LZW), ZIP, or bzip2, achieve their gains through dictionary-building or transforms, but nonetheless rely on an implicit statistical model of the domain.[1]

---

[1]They can even be used to create explicit predictive models [Delétang et al. (2024)].

In model-driven compression, the message is encoded sequentially, and for each symbol the model makes a probabilistic prediction based on the context of the prior symbols to help the encoder efficiently allocate bits to potential outcomes. To convert the predictive model into a compression algorithm, the standard technique is to pair the model with arithmetic coding [Pasco (1976); Rissanen (1976); Guazzo (1980)]. Arithmetic coding represents an entire message as a subinterval of $[0, 1)$, successively narrowing the interval according to the predicted probabilities of each symbol. More probable symbols shrink the interval less and thus yield shorter average descriptions, while less probable symbols shrink it more and thus require more bits. Unlike Huffman coding [Huffman (1952)] (another commonly used technique), arithmetic coding adapts particularly well to changing and context-dependent probabilities for each symbol. If the model closely reflects the true distribution of the data, arithmetic coding yields compression rates approaching the information-theoretic limit. However, it is extremely sensitive numerically and vulnerable to cascading errors.

Model-driven lossless compression has a long history, arguably going back to Shannon (1948), where Shannon tallies frequencies of characters in English, building first, second, and third order Markov model predictions. The arithmetic coding approach for model-driven compression was discussed by Cleary & Witten (1984) and further developed with a number of statistical or learned predictive models across many domains. Many of these models focus on deriving the prediction model from only the previously seen encoded symbols. In Schmidhuber & Heil (1996), it is clarified that "offline" models are those trained on a separate files and model parameters are shared among all machines responsible for encoding and decoding. In contrast "online" models use the current file to update predictions. Schmidhuber & Heil (1996) use offline neural networks and get competitive compression ratios. Many other works since, Knoll (2025); Cox (2016); Goyal et al. (2018); Bellard (2019); Liu et al. (2019) have used LSTMs and other recurrent neural networks as predictive models. Transformers were used as the predictive model in Bellard (2019; 2021); Mao et al. (2022).

This general arithmetic coding-based lossless compression technique, particularly when paired with modern neural network-driven predictive models, has been shown to have significant promise in numerous domains beyond text compression. These domains include lossless image compression Toderici et al. (2016); Schiopu et al. (2018); Mentzer et al. (2019); Rhee et al. (2022); Chen et al. (2024), compression of large numerical datasets such as time-series power data Ma et al. (2022), and neural network checkpoints Kim & Belyaev (2025).

The incredible success of modern neural networks, particularly transformers, for natural language processing has led to increased interest in using the model-driven approach to create more powerful and context-adaptive codes for natural language compression. Recent work by Delétang et al. (2024) shows that offline model-driven compression using modern models such as Llama 2 or Chinchilla with arithmetic coding can deliver significant improvement over state-of-the-art lossless compression algorithms across domains including text and vision. Concurrently, LLM-driven text compression tools such as LLMZip [Valmeekam et al. (2023)] and llama-zip [Buzanis (2024)] were introduced to take advantage of the capabilities of these advanced models.

## 1.2 LLM Non-Determinism and Prediction Mismatch

Despite its promise, LLM-driven compression faces serious practical obstacles. For instance, the LLM inference pipeline must run for each token during the encoding and decoding steps, which can make the process prohibitively slow, since large language models are often computationally expensive to execute. This also requires the model, which may contain many gigabytes of parameters, to be stored, thus adding a large overhead cost in memory as well. Recent work has also been done to address concerns about the computational performance of LLM-driven compression, such as Mittu et al. (2024) on improving speeds for LLMZip.

Another significant challenge, which we call *prediction mismatch*, arises when compressed data is transmitted between an encoder and decoder running on different machines. As noted in Witten et al. (1987), arithmetic coding with adaptive probability models, "It must be possible for the decoder to produce exactly the same probability distribution in the same context". Achieving this is difficult with modern machine learning models due to *non-determinism*.

Non-determinism in the setting of machine learning and scientific computing means that multiple runs of the same program with identical inputs (and identical random seeds) can produce different outputs [Cooper et al. (2022); Semmelrock et al. (2025)]. One source of non-determinism occurs

in GPU hardware: floating point operations which are performed in a different order may result in different outcomes due to rounding. These small numerical deviations, in a full inference pipeline run, can cascade into large differences in what a model predicts [Shanmugavelu et al. (2025); Chen et al. (2022)]. GPU libraries, like CUDA and cuDNN, state specifically in their documentation that they do not guarantee determinism or reproducibility in many circumstances, such as when different versions or architectures are used [NVIDIA Corporation (2025a;b)]. The effects of non-determinism in CUDA is studied in Eryilmaz et al. (2024) where they note that non-determinism is likely to remain in CUDA because of the runtime benefits CUDA gains through using parallelism. Non-determinism in GPUs have also been examined and explored by Morin & Willetts (2020). Coakley et al. (2022) and Atil et al. (2025) examined the issue of non-determinism experimentally, finding significant variability, and Schlögl et al. (2023) study its causes.

Applying arithmetic coding directly under these conditions is usually immediately fatal: even subtle differences in the encoder and decoder probability distributions can result in an incorrectly decoded token, which then cascades to the rest of the message as it changes the context of subsequent tokens.

If the mismatch between the encoder and decoder distributions is arbitrary, recovery is impossible, as the decoder predictions yield no information about the encoder predictions. However, if the mismatch is known to be small, the encoder and decoder can exploit this closeness to reach exact agreement on a third probability distribution. This robustness incurs a cost in compression efficiency: the encoder will generally have to send extra information to ensure agreement, and the agreed probability distribution may be less accurate than the original predictions. We refer to the problem of constructing a shared distribution while minimizing the cost as *probability matching*.

Concurrent work [Hu & Tang (2026)] addresses a closely related variant of mismatch-tolerant coding by proposing an alternative algorithm based on a Huffman coding-like approach.

Recent work has also explored the possibility of addressing non-determinism at its source by designing deterministic neural network backends, motivated by the need for researchers to share reproducible results [He & Lab (2025), Yuan et al. (2025)], though at a cost in performance. Mitigation of non-determinism can be complementary with our approach by reducing mismatches so that they fall within the tolerance limits of a robust coding algorithm.

## 1.3 CONTRIBUTIONS

This work introduces the problem of robust coding for prediction mismatch in model-driven lossless compression and proposes *PMATIC* (Probability-Matched Interval Coding) to address it. PMATIC is designed to convert any predictive model into a compression algorithm which is robust to bounded prediction mismatch, and to be a drop-in replacement for arithmetic coding in model-driven compression. This work also shows the following results:

- *Theory:* We prove that PMATIC guarantees correct decoding under a simple and general model of bounded prediction mismatch (Section 2.1), and give theoretical bounds on the cost incurred to ensure this robustness.

- *Practice:* We demonstrate experimentally that LLM-driven compression using PMATIC achieves compression ratios significantly better than current standard methods, while remaining robust to prediction mismatch.

In Section 5, we validate our approach by applying PMATIC on text data in the presence of both real and synthetic prediction mismatch, and give the first proof of concept that non-determinism in model-driven compression can be addressed via robust coding algorithms.

## 2 PROBLEM STATEMENT

Consider the case where an encoder and decoder are using the same model (such as a specific LLM with the same weights) to compute next-token probabilities over an input string $x = x(1)x(2)\ldots x(n)$ whose entries $x(i)$ are taken from a finite alphabet $\mathcal{A}$ of possible symbols. We use the following notation: the $i$-symbol prefix of $x$ is denoted as $x^i := x(1)\ldots x(i)$; the set of all finite strings drawn from the alphabet $\mathcal{A}$ is denoted as $\mathcal{A}^* := \bigcup_{i \geq 0} \mathcal{A}^i$.

Typically, an LLM computes its next-token probabilities by computing a real-valued (or, rather, floating-point valued) weight, called a *logit*, for each outcome and then applying the *softmax* function to the vector of logits.[2] Let functions $M^{\text{Enc}}, M^{\text{Dec}} : \mathcal{A}^* \to \mathbb{R}^{\mathcal{A}}$ take a string of symbols (the context) and return a logit value for each symbol in the alphabet, representing what happens when the model is run for inference at the encoder and decoder ends, respectively. We denote the predictions of $M^{\text{Enc}}, M^{\text{Dec}}$ for token $i$ as logit vectors

$$u(i) := M^{\text{Enc}}(x^{i-1}) \quad \text{and} \quad v(i) := M^{\text{Dec}}(x^{i-1}) \tag{1}$$

which respectively induce probability vectors $p(i) = \text{softmax}(u(i))$ and $q(i) = \text{softmax}(v(i))$, i.e. for any $i \in [n]$ and $k \in \mathcal{A}$,

$$p(i)_k = \text{softmax}(u(i))_k := \frac{e^{u(i)_k}}{\sum_{j \in \mathcal{A}} e^{u(i)_j}} \quad \text{and} \quad q(i)_k = \text{softmax}(v(i))_k := \frac{e^{v(i)_k}}{\sum_{j \in \mathcal{A}} e^{v(i)_j}}. \tag{2}$$

When the token number $i$ is fixed, we may drop it from the notation for clarity, so that the encoder and decoder return logit vectors $u, v$ which induce probability distributions $p, q$ respectively.

We denote the encoding and decoding algorithms (also called compressing and decompressing, respectively) as functions whose operation depends on an LLM model ($M^{\text{Enc}}$ and $M^{\text{Dec}}$ respectively) as well as on more traditional inputs. Specifically, the encoder takes LLM $M^{\text{Enc}}$ and input token string $x$ and returns a bitstring $b$ which is the encoded input. Then, the decoder takes $b$ and its own LLM $M^{\text{Dec}}$ and returns a decoded string $\hat{x}$:

$$\text{Enc}(M^{\text{Enc}}; x) = b \quad \text{and} \quad \text{Dec}(M^{\text{Dec}}; b) = \hat{x}. \tag{3}$$

Given a constraint on the difference between $M^{\text{Enc}}$ and $M^{\text{Dec}}$'s outputs on any given context, we say that the algorithm is *mismatch-tolerant* with respect to that constraint if, for all $M^{\text{Enc}}, M^{\text{Dec}}$ that satisfy the constraint: $\text{Dec}(M^{\text{Dec}}; \text{Enc}(M^{\text{Enc}}; x)) = x$ for all $x$. The goal is to design algorithms which can tolerate a given amount of mismatch between the encoder and decoder probability distributions while minimizing the cost in compression efficiency.

## 2.1 The Bounded Prediction Mismatch Setting

Since the encoder and decoder are using the same LLM on the same inputs, it is reasonable to assume that they obtain logits whose difference is bounded by some reasonably small $\varepsilon > 0$. A natural choice for this is to assume that their difference has bounded $L_\infty$ norm (i.e. elementwise): $\|u - v\|_\infty := \max_{k \in \mathcal{A}} |u_k - v_k| \le \varepsilon$. We first define a measure of difference between two probability distributions:

**Definition 1.** *The conditional total variation distance ($d_{\text{CTV}}$) between two probability distributions $p, q$ on an alphabet $\mathcal{A}$ is defined as the maximum total variation distance ($d_{\text{TV}}$) of $p$ and $q$ after conditioning on some (nonempty) $S \subseteq \mathcal{A}$, i.e.*

$$d_{\text{CTV}}(p, q) := \max_{\emptyset \ne S \subseteq \mathcal{A}} d_{\text{TV}}(p(\cdot|S), q(\cdot|S)) \tag{4}$$

*where $p(\cdot|S)$ and $q(\cdot|S)$ are, respectively, $p$ and $q$ conditioned on the outcome being in $S$.*

Note that there is no divide-by-zero issue with conditioning on any (nonempty) $S$ since all probabilities are induced via the softmax function and hence strictly positive. Bounded prediction mismatch then bounds conditional TV distance:

**Proposition 1.** *If $u, v$ induce probability distributions $p = \text{softmax}(u)$ and $q = \text{softmax}(v)$ over $\mathcal{A}$, and $\|u - v\|_\infty \le \varepsilon$, then $d_{\text{CTV}}(p, q) \le \frac{\varepsilon}{2}$.*

*Proof.* Using the definition of TV distance, the conditional TV distance can be rewritten as

$$d_{\text{CTV}}(p, q) = \max_{\substack{\emptyset \ne S \subseteq \mathcal{A} \\ S^* \subseteq S}} |p(S^*|S) - q(S^*|S)| \tag{5}$$

$$\implies \max_{\|u-v\|_\infty \le \varepsilon} d_{\text{CTV}}(p, q) = \max_{\|u-v\|_\infty \le \varepsilon} \max_{\substack{\emptyset \ne S \subseteq \mathcal{A} \\ S^* \subseteq S}} |p(S^*|S) - q(S^*|S)| \tag{6}$$

$$= \max_{\substack{\emptyset \ne S \subseteq \mathcal{A} \\ S^* \subseteq S}} \max_{\|u-v\|_\infty \le \varepsilon} |p(S^*|S) - q(S^*|S)| \tag{7}$$

---

[2]For simplicity we use the standard softmax with a 'temperature' parameter of 1.

So, if $\max\limits_{\|u-v\|_\infty \leq \varepsilon} |p(S^*|S) - q(S^*|S)| \leq \frac{\varepsilon}{2}$ for all $S^* \subseteq S \subseteq \mathcal{A}$, then $\max\limits_{\|u-v\|_\infty \leq \varepsilon} d_{\mathrm{CTV}}(p,q) \leq \frac{\varepsilon}{2}$.

In other words, the conditional TV distance between $p, q$ induced by $\|u-v\|_\infty \leq \varepsilon$ can be bounded by first fixing $S^* \subseteq S \subseteq \mathcal{A}$ and then bounding $|p(S^*|S) - q(S^*|S)|$ over all $p, q$ whose logits are within $\varepsilon$ of each other in $L_\infty$ distance. We assume WLOG that the $u, v$ maximizing $|p(S^*|S) - q(S^*|S)|$ has $q(S^*|S) > p(S^*|S)$ (otherwise $S^*$ can be changed into $S \backslash S^*$), so the goal is to maximize $q(S^*|S) - p(S^*|S)$ given the logit $L_\infty$ bound. This is achieved by letting

$$v_k = u_k + \varepsilon \text{ for } k \in S^* \quad \text{and} \quad u_k - \varepsilon \text{ for } k \notin S^* \tag{8}$$

Let $p^* := p(S^*|S)$ and $q^* := q(S^*|S)$, which are both scalars in $[0,1]$. Then, given (8),

$$q^* = \frac{\sum_{k\in S^*} e^{u_k+\varepsilon}}{\sum_{k\in S^*} e^{u_k+\varepsilon} + \sum_{k\in S\backslash S^*} e^{u_k-\varepsilon}} = \frac{e^\varepsilon p^*}{e^\varepsilon p^* + e^{-\varepsilon}(1-p^*)} \tag{9}$$

$$\implies q^* - p^* \leq \max_{p^*\in[0,1]} \left( \frac{e^\varepsilon p^*}{e^\varepsilon p^* + e^{-\varepsilon}(1-p^*)} - p^* \right) = \tanh\left(\frac{\varepsilon}{2}\right) \leq \frac{\varepsilon}{2}. \tag{10}$$

Tracing this bound back to the conditional TV distance concludes the proof.[3] $\qquad\qquad\square$

# 3 THE PMATIC ALGORITHM

The Probability Matching Interval Coding (PMATIC) algorithm addresses prediction mismatch by ensuring that the encoder and decoder use a common probability distribution for each token. The first step of PMATIC is to convert the input token string into a bitstring using a dictionary that associates each token with a length $\ell := \lceil \log_2(|\mathcal{A}|) \rceil$ bitstring; we call this bitstring the token's *longform*[4] and each bit in it is a *token bit*. PMATIC encodes these tokens bits with arithmetic coding using next-bit conditional probabilities derived from the token's encoder prediction vector. At each step, the next-bit prediction is a scalar in $[0,1]$ giving the probability that the token bit equals $1$. The key idea is to divide the interval $[0,1]$ into a set of *bins* (disjoint equal-length intervals which cover $[0,1)$). Then, instead of using their exact predictions, the encoder and decoder use either the center of the bin their predictions fall into or the nearest boundary between two bins; which one to use is decided by the encoder and communicated to the decoder by use of auxiliary 'helper' bits, which are also sent via arithmetic coding. This procedure can be viewed as quantizing the probability of each token bit.[5]

For any token $x_i$ in the message, we denote its longform by $b_i := b_i(1) \ldots b_i(\ell)$ and define the following parameters and notation:

- $\delta > 0$, which represents the amount of prediction mismatch (per bit) which the algorithm can tolerate, as measured by conditional TV distance.

- $r > 0$ is the radius of the quantization bins (so the width of a bin is $2r$); $r$ will be chosen to maximize performance given $\delta$. We will assume that $r = 1/(2m)$ for some integer $m$; in practice this entails rounding $r$ up or down slightly to the nearest such value. We also require the constraint that $r > 2\delta$.

- Let $h(p) := p \log\left(\frac{1}{p}\right) + (1-p) \log\left(\frac{1}{1-p}\right)$ be the binary entropy function (the entropy of a Bernoulli random variable with probability $p$), and $H(p)$ be the more general entropy function for a probability vector $p$ over a finite set.

- Let $d_{\mathrm{KL}}(p\|q) := p \log\left(\frac{p}{q}\right) + (1-p) \log\left(\frac{1-p}{1-q}\right)$ be the binary Kullback Liebler divergence (the divergence between Bernoulli random variables with probabilities $p, q$).

- Let $S_{b_i^{j-1}} := \{a \in \{0,1\}^\ell : a^{j-1} = b_i^{j-1}\}$ be the longforms whose first $j-1$ bits match $b_i$.

---

[3]For a full explanation of the maximization step in (10), see Appendix A.1.

[4]This term is also used in Hu & Tang (2026) to describe a similar construction.

[5]Note that when we say that a bit is encoded or decoded using a probability value $p$, we mean that the corresponding arithmetic coding step is performed using the probability distribution $(1-p, p)$ to set the intervals.

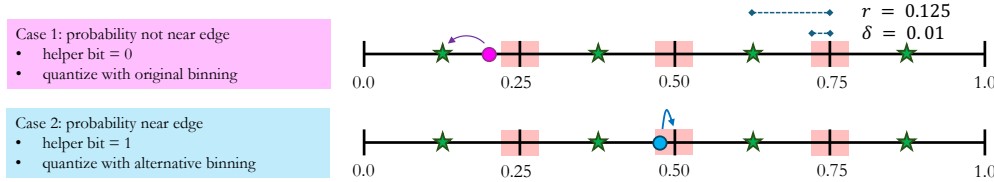

Figure 1: Examples of PMATIC helper-bit and quantization logic for two cases, one where the helper bit is 0 and one where the helper bit is 1.

## 3.1 The PMATIC Encoder

Consider encoding the $j$th bit, $b_i(j)$, of token $x_i$. Let the encoder and decoder prediction vectors for token $i$ be, respectively, $p(i) := \text{softmax}(\mathbf{M}^{\text{Enc}}(x^{i-1}))$ and $q(i) := \text{softmax}(\mathbf{M}^{\text{Dec}}(x^{i-1}))$. The predictions for the $j$th bit $b_i(j)$ for the encoder and decoder, conditional on the prior bits in $b_i$, are

$$p_i(j) := \mathbb{P}_{p(i)}[b_i(j) = 1 \,|\, S_{b_i^{j-1}}] \quad \text{and} \quad q_i(j) := \mathbb{P}_{q(i)}[b_i(j) = 1 \,|\, S_{b_i^{j-1}}] \tag{11}$$

These can be computed directly using $p(i)$ (or $q(i)$) by setting all values outside of $S_{b_i^{j-1}}$ to 0 and renormalizing to get the conditional probability distribution.

The interval $[0, 1]$ is then split into radius-$r$ intervals, which we call *bins*, $I_1, I_2, \dots I_m$, where $m = 1/(2r)$ (which, as assumed above, is an integer) and $I_k = [2r(k-1), 2rk]$. The center of bin $I_k$ is therefore $c_k := 2r(k-1) + r$, and we denote the $\delta$-*interior* of $I_k$ (the set of points in $I_k$ at least $\delta$ away from any point outside $I_k$) by

$$I_k^\delta = \begin{cases} [0, 2r - \delta] & \text{if } k = 1 \\ [2r(m-1) + \delta, 1] & \text{if } k = m \\ [2r(k-1) + \delta, 2rk - \delta] & \text{if } k \neq 1, m \end{cases} \tag{12}$$

$I_1^\delta, I_m^\delta$ get special definition as they each have an edge next to the edge of $[0, 1]$.

Note that if $p_i(j) \in I_k^\delta$ and $|p_i(j) - q_i(j)| \leq \delta$, then $q_i(j) \in I_k$, and that if $p_i(j) \notin I_k^\delta$ for all $k$, then there is instead a unique integer $k \neq 1, m$ for which $|2rk - p_i(j)| < \delta$.

In addition to the token bit $b_i(j)$, PMATIC encodes (prior to the token bit) a *helper bit*

$$b_i'(j) = \begin{cases} 0 & \text{if } b_i(j) \in I_k^\delta \text{ for some } k \\ 1 & \text{otherwise} \end{cases} \tag{13}$$

This is encoded with arithmetic coding using probabilities $p' := \delta/r$ for the helper bit and

$$\hat{p}_i(j) = \begin{cases} c_k = 2r(k-1) + r & \text{if } p_i(j) \in I_k^\delta \\ 2rk \text{ for the integer } k \text{ s.t. } |2rk - p_i(j)| < \delta & \text{otherwise} \end{cases} \tag{14}$$

for the token bit. The probability $\hat{p}_i(j)$ is intended as the common probability of token bit $j$ that both the encoder and decoder agree to use. See Figure 1 for an example.

The intuition for the helper bits is that when $p_i(j) \in I_k^\delta$, the encoder knows that the decoder's probability $q_i(j) \in I_k$ (the same bin), so the encoder quantizes to the bin center and tells the decoder to do the same by sending the helper bit $b_i'(j) = 0$. If $p_i(j)$ is not in the $\delta$-interior of its bin, the encoder no longer knows that the decoder probability lies in the same bin. However, in this case both probabilities must be near the same boundary point between two bins, so the encoder quantizes to the nearest boundary point and tells the decoder to do the same by sending the helper bit $b_i'(j) = 1$. In either case, they agree on the probability to use for encoding and decoding. The probability of being in the $\delta$-interior of a bin is $\approx \delta/r$, which is very small if $r \gg \delta$. This gives the helper bits low entropy and hence makes them also highly compressible via arithmetic coding.

To summarize, given a token $x_i$ and context $x^{i-1}$, the PMATIC encoder does the following:

1. Computes $p(i) = \text{softmax}(\mathbf{M}^{\text{Enc}}(x^{i-1}))$, gets the longform $b_i$ corresponding to $x_i$, and computes the conditional next-bit probabilities $p_i(1), \dots, p_i(\ell) \in [0, 1]$.

2. Computes for each $b_i(j)$ the helper bit $b_i'(j)$ and quantized probability $\hat{p}_i(j)$ ((13), (14)).

3. Encodes the bitstring $b_i'(1)b_i(1) \dots b_i'(\ell)b_i(\ell)$ using arithmetic coding, where the encoding probability for helper bit $b_i'(j)$ is $p' = \delta/r$ and for token bit $b_i(j)$ is $\hat{p}_i(j)$.

An example of the encoding process for one token is given in Appendix A.2.

## 3.2 THE PMATIC DECODER

The PMATIC decoder takes the encoded message $y$ and decodes it sequentially in pairs of bits. Each pair consists of a helper bit and a token bit; the helper bit is decoded first using $p' = \delta/r$ as the probability (since helper bits are always encoded using this probability), and determines the quantized probability to use. Analogous to the encoder next-bit prediction, let the decoder next-bit prediction for bit $j$ of token $i$ be denoted $q_i(j)$. Then the decoder decodes the token bit using the quantized probability:

$$\hat{q}_i(j) = \begin{cases} c_k \text{ for } k \text{ s.t. } q_i(j) \in I_k & \text{if } b_i'(j) = 0 \\ 2rk \text{ for } k \in \{1, \dots, m-1\} \text{ s.t. } |2rk - q_i(j)| \text{ is minimized} & \text{if } b_i'(j) = 1 \end{cases} \quad (15)$$

After decoding all the bits, the helper bits are discarded and the token bits are converted back into the token using the longform dictionary. The token is then added to the context of the predictive model and predictions are generated for the next token. PMATIC is successful when $\hat{q}_i(j)$ is the always the same as $\hat{p}_i(j)$. This is discussed in greater detail in the next section.

## 4 ANALYSIS

We wish to: (i) show PMATIC ensures correctness if the conditional TV distance between the encoder and decoder token predictions is at most $\delta$; (ii) show (expected) theoretical performance bounds. Since we compress in bits, logarithms are base-2 unless noted otherwise.

## 4.1 CORRECTNESS

**Theorem 1.** *If $d_{\text{CTV}}(p(i), q(i)) \leq \delta$, then $\hat{q}_i(j) = \hat{p}_i(j)$ for all $j$ (i.e. the encoder and decoder will agree on the quantized probabilities for all bits corresponding to token $x_i$).*

*Proof.* Consider bit $j$ of token $i$; without loss of generality we can assume that all previous bits in $i$ and all previous tokens were decoded correctly (since, if any bits are incorrectly decoded, there must be a first one). Since $d_{\text{CTV}}(p(i), q(i)) \leq \delta$, we know that $|p_i(j) - q_i(j)| \leq \delta$ (since $p_i(j), q_i(j)$ are derived by conditioning $p(i), q(i)$ on the set $S_{b_i^{j-1}}$).

First, the decoder will decode the helper bit $b_i'(j)$ using the probability $\delta/r$. Since $\delta/r$ is fixed and used for all helper bits, the encoder and decoder probabilities match and the helper bit is decoded correctly. Now we consider two cases: $b_i'(j) = 0$, and $b_i'(j) = 1$.

If $b_i'(j) = 0$, this means that computed encoder next-bit predictor $p_i(j)$ falls in the $\delta$-interior $I_k^\delta$ of some bin; since $|p_i(j) - q_i(j)| \leq \delta$, this means $q_i(j) \in I_k$ (not necessarily the $\delta$-interior, just the bin itself). Thus, since both the encoder and decoder quantize to the center $c_k$ of the bin, we have $\hat{p}_i(j) = \hat{q}_i(j) = c_k$ and the token bit is encoded correctly.

If $b_i'(j) = 1$, then there is some $k \in \{1, \dots, m-1\}$ such that $|p_i(j) - 2rk| \leq \delta$ (note that $2rk$ here is the boundary between two bins). Then, since we set $r > 2\delta$ and bins have width $2r$:

$$|p_i(j) - 2rk| \leq \delta \implies |q_i(j) - 2rk| \leq 2\delta \quad (16)$$

$$\implies |q_i(j) - 2rk'| \geq 2r - 2\delta > 2\delta \text{ for any integer } k' \neq k \quad (17)$$

$$\implies \hat{q}_i(j) = 2rk = \hat{p}_i(j). \quad (18)$$

Thus, in either case, the encoder and decoder agree on the next-bit probability for $b_i(j)$ and the decoder will decode the bit and update the arithmetic code interval correctly. $\square$

Note that, by Theorem 1 and Proposition 1, if the LLM has logits that differ by at most $\varepsilon$ between the encoder and decoder, then PMATIC using $\delta = \varepsilon/2$ guarantees correctness.

## 4.2 COMPRESSION LOSS

For the compression performance analysis, we make the following simplifying assumptions:

- The encoder's next-token probabilities are the true probabilities, so the expected length increase per bit $b_i(j)$ is $d_{\text{KL}}(p_i(j)\|\hat{p}_i(j))$ (Cover & Thomas, 2006, Thm 5.4.3).

- Within each individual bin, the encoder next-bit probability is roughly uniformly distributed, so probability of being within $\delta$ of a bin boundary is $\approx \delta/r$ (this is an approximation since we ignore the fact that the first and last bins have only one relevant boundary each). This is a 'worst-case non-adversarial' assumption: next-bit probabilities do not disproportionately cluster near bin boundaries, but we otherwise know nothing about them.

We consider the *compression loss* of PMATIC over traditional (non-mismatch-tolerant) arithmetic coding, i.e. the extra message length incurred by PMATIC in order to tolerate a conditional TV distance bound of $\delta$ with a bin width of $r$. This loss comes from two sources:

1. *Helper bit encoding:* Since helper bits are assumed to be Bernoulli with parameter $\delta/r$, the expected extra encoding length per helper bit is the binary entropy $h(\delta/r) = \frac{\delta}{r}\log\left(\frac{r}{\delta}\right) + \left(\frac{r-\delta}{r}\right)\log\left(\frac{r}{r-\delta}\right)$. If $r \gg \delta$, the first term dominates and the entropy is $\approx \frac{\delta}{r}\log\left(\frac{r}{\delta}\right)$.

2. *Quantization loss:* The quantized probability $\hat{p}_i(j)$ is different than the true probability $p_i(j)$, incurring a quantization loss of $d_{\text{KL}}(p_i(j)\|\hat{p}_i(j))$.
   Since $r \leq \hat{p}_i(j) \leq 1 - r$ and $|p_i(j) - \hat{p}_i(j)| \leq r$, this satisfies $d_{\text{KL}}(p_i(j)\|\hat{p}_i(j)) \leq 2\log(e)r$, since KL divergence is bounded above by $\chi^2$ divergence (Polyanskiy & Wu, 2025, Ch. 7):

$$d_{\text{KL}}(p_i(j)\|\hat{p}_i(j)) \leq \log(e)\frac{(p_i(j) - \hat{p}_i(j))^2}{\hat{p}_i(j)(1 - \hat{p}_i(j))} \leq 2\log(e)r \tag{19}$$

since the numerator is $\leq r^2$ and the denominator is $\geq (1/2)r$.

Note that these losses respond in opposite directions when the bin radius $r$ is increased: larger bins mean lower helper bit entropy but a bigger difference between the true probability and the quantized probability. This means that setting $r$ to balance the loss terms gives an approximate minimizer of the objective function: any other $r' \neq r$ will make one of the loss terms larger, so balancing the loss terms incurs a total loss of at most 2 times the optimal. This is achieved (approximately) with

$$2\log(e)r = \frac{\delta}{r}\log\left(\frac{r}{\delta}\right) \implies r \approx \frac{\sqrt{\delta\log\left(\frac{1}{\delta}\right)}}{\sqrt{2\log e}} \tag{20}$$

and yields a total loss on the order of $O\left(\sqrt{\delta\log\left(\frac{1}{\delta}\right)}\right)$.

## 5 EXPERIMENTS

We test PMATIC in model-driven text compression algorithms driven by: LLaMA 3.1 8B (4-bit quantized) [Grattafiori et al. (2024)], Mistral 7B v0.1 (3-bit quantized) and Qwen 2.5 Instruct 7B (3-bit quantized), whose alphabets have respective sizes of $128,256$ tokens, $32,000$ tokens, and $151,643$ tokens. We test PMATIC with three robustness settings $\delta$, with bin width $r$ chosen according to (20), rounded to a nearby reciprocal of an integer (in order to get fully uniform bin sizes): (i) $\delta = 0.00001$ and $r = 0.005$; (ii) $\delta = 0.001$ and $r = 0.05$; and (iii) $\delta = 0.1$ and $r = 0.125$.

We compare PMATIC against (i) non-robust model-driven text compression (with the same models) using standard arithmetic coding, and (ii) a set of traditional or modern text compression algorithms, including gzip (widely used in practice) and CMIX (known for achieving state-of-the-art compression ratios). Comparing PMATIC against standard arithmetic coding within the same model-driven compression pipelines shows the 'robustness overhead' cost from PMATIC (how much additional information is sent to ensure robustness of the chosen level)[6], while the comparisons to traditional

---

[6]While there are existing benchmarks for model-driven compression [Valmeekam et al. (2023), Mittu et al. (2024)], the effectiveness of model-driven compression depends heavily on the model and setting used. Thus, to isolate the robustness-efficiency tradeoff produced by PMATIC, we compare PMATIC with standard arithmetic coding within otherwise-identical pipelines.

text compression algorithms validates the effectiveness of the model-driven compression pipeline even with the additional overhead from ensuring robustness.

We run our experiment on several datasets: (i) the first (nearly) 10 MB of the enwik8 benchmark Hutter (2006), a collection of Wikipedia articles from 2006; (ii) 1000 randomly selected articles from Wikipedia; (iii) *Hamlet* by Shakespeare and (iv) *Emma* by Austen, in English; (v) *Candide* by Voltaire in French; and (iv) *Dream of the Red Chamber* (红楼梦) by Cao Xuewin, in Chinese. Additional implementation details are in Appendix A.3.

## 5.1  RESULTS: COMPRESSION RATIO

| | | Enwik8 | Wikipedia | Hamlet | Emma | Candide *French* | 红楼梦 *Chinese* |
|---|---|---|---|---|---|---|---|
| **LLM-based compression (with and without PMATIC)** | | | | | | | |
| Meta LLaMA 3.1 | no PMATIC | 0.0780 | 0.0700 | 0.0878 | 0.0606 | 0.1024 | – |
| | $\delta = 10^{-5}, r = 0.005$ | 0.0847 | 0.0878 | 0.0952 | 0.0660 | 0.1102 | – |
| | $\delta = 10^{-3}, r = 0.05$ | 0.1353 | 0.1330 | 0.1514 | 0.1099 | 0.1683 | – |
| | $\delta = 10^{-2}, r = 0.125$ | 0.2492 | 0.2085 | 0.2772 | 0.2113 | 0.2971 | – |
| Mistral 7B | no PMATIC | 0.0867 | 0.0737 | 0.1501 | 0.1066 | – | – |
| | $\delta = 10^{-5}, r = 0.005$ | 0.0940 | 0.0794 | 0.1587 | 0.1134 | – | – |
| | $\delta = 10^{-3}, r = 0.05$ | 0.1481 | 0.1198 | 0.2167 | 0.1595 | – | – |
| | $\delta = 10^{-2}, r = 0.125$ | 0.2699 | 0.2106 | 0.3447 | 0.2610 | – | – |
| Qwen2.5 7B Instruct | no PMATIC | 0.0881 | 0.0824 | 0.1102 | 0.1183 | 0.1150 | 0.1268 |
| | $\delta = 10^{-5}, r = 0.005$ | 0.0951 | 0.0880 | 0.1177 | 0.1253 | 0.1231 | 0.1345 |
| | $\delta = 10^{-3}, r = 0.05$ | 0.1501 | 0.1315 | 0.1755 | 0.1738 | 0.1831 | 0.1879 |
| | $\delta = 10^{-2}, r = 0.125$ | 0.2751 | 0.2297 | 0.3067 | 0.2825 | 0.3171 | 0.3073 |
| **Standard baselines** | | | | | | | |
| cmix | | 0.3558 | 0.3644 | 0.3865 | 0.3797 | 0.3709 | 0.3824 |
| brotli | level 11 | 0.3524 | 0.3546 | 0.4361 | 0.3918 | 0.4442 | 0.4733 |
| bzip2 | level 9 | 0.4537 | 0.4605 | 0.4636 | 0.4539 | 0.4494 | 0.4727 |
| xz | level 9 | 0.4647 | 0.4904 | 0.5111 | 0.4973 | 0.4866 | 0.5192 |
| gzip | level 9 | 0.4601 | 0.4759 | 0.5007 | 0.4852 | 0.4768 | 0.5305 |
| zstd | level 22 | 0.4676 | 0.4773 | 0.5034 | 0.4882 | 0.4851 | 0.5544 |

Table 1: Compression ratio ($\frac{\text{compressed file size}}{\text{uncompressed file size}}$, lower is better) across different models, PMATIC parameters, and datasets. PMATIC overhead (cost of gaining mismatch robustness through PMATIC) for each parameter setting can be computed by subtracting the corresponding 'no PMATIC' compression ratio from the PMATIC compression ratio corresponding to that robustness setting.

Compression efficiency results are given in Table 1 (where compression ratios are given as $\frac{\text{compressed file size}}{\text{uncompressed file size}}$, following the convention of Delétang et al. (2024)), and show that PMATIC provides robustness to numerical deviations while keeping favorable compression ratios over traditional algorithms, across the models and settings we tested. Even in the most robust setting, compression with PMATIC still achieves significantly better compression rates than traditional compression.

We also studied whether the 'worst-case non-adversarial' assumption that the next-bit probabilities have a $\approx \delta/r$ chance of falling within $\delta$ of a bin boundary (and thus setting the corresponding helper bit to 1) was accurate. Table 2 compares the expected fraction of helper bits set to 1 (under the uniformity assumption) to the actual fraction. The last column gives the fraction of the size of the compressed files dedicated to helper bits under each robustness setting, showing how much improvement might be obtained by improving the compression of the helper bits alone. The results show that helper bits are considerably less likely to be 1 in practice than the uniformity assumption indicates. This is because the next bit is often essentially certain (probability close to 0 or 1, which do not fall near bin boundaries), particularly when earlier bits have narrowed down possible next token. Because PMATIC relies on low helper-bit entropy to efficiently communicate them, a lower fraction of helper bits being 1 is better; in our implementation, the helper bits were encoded with arithmetic coding using the uniformity-assumption probability of $\delta/r$, but these results indicate that using more accurate helper bit probabilities can yield significant improvement.

| Parameter settings | Expected helper 1 fraction | Measured helper 1 fraction | Helper bit cost |
|---|---|---|---|
| no PMATIC | 0 | 0 | 0 |
| $\delta = 10^{-5}$, $r = 0.005$ | 0.002 | 0.00051 | 0.04594 |
| $\delta = 10^{-3}$, $r = 0.05$ | 0.02 | 0.00368 | 0.18947 |
| $\delta = 10^{-2}$, $r = 0.125$ | 0.08 | 0.01145 | 0.34073 |

Table 2: Helper bit behavior averaged across different PMATIC robustness parameter settings. *Expected helper* 1 *fraction* is $\delta/r$, approximately the fraction of helper bits which would be set to 1 if next-bit probabilities are uniformly distributed; *Measured helper* 1 *fraction* is the actual fraction of helper bits that were 1, across all models and datasets; and *Helper bit cost* is the fraction of the size of the compressed files dedicated to storing helper bits. The results indicate that in practice, next-bit probabilities are much *less* likely to fall within $\delta$ of a bin boundary than the uniformity assumption implies, and that accounting for this could make PMATIC considerably more efficient.

## 5.2 Results: Robustness to Synthetic and Real Non-Determinism

To test the theoretical correctness of PMATIC (Theorem 1), we added IID synthetic noise from $[-2\delta, 2\delta]$ (conforming to the theoretical guarantee in Proposition 1) to each predicted logit before decoding. As expected, all files were decoded successfully.

We also ran a smaller number of tests on PMATIC under real non-determinism resulting from encoding and decoding the output on two different machines using Meta Llama 3.1. In these tests, the encoding and decoding were performed on two different Apple laptops (MacBook Pro) using, respectively, the Apple M2 Pro chipset with a 16 core GPU and the Apple M4 Max chipset with a 32 core GPU. This test was conducted on 100 Wikipedia articles and the first 250 KB of *Emma*, split into 50 files each with size 5 KB (both are a subset of the files used in the compression ratio tests). No files were correctly decoded when using arithmetic coding (no PMATIC) or PMATIC with $\delta = 0.001$. However, when tested with parameter setting $\delta = 0.01$, all files were correctly decoded. This matched preliminary explorations indicating that $\delta = 0.01$ is a reasonable estimate for the upper bound on the discrepancy in Llama 3.1 outputs between the two laptops. See Appendix A.4 for more.

## 6 Future Work

While this work has focused on the initial implementation and validation of PMATIC using text models and datasets, we view the application of PMATIC to compression in other domains, such as images, to be a natural extension.

Furthermore, PMATIC is designed for the bounded prediction mismatch setting; however, logit mismatches arising from non-determinism may obey stochastic bounds rather than strict upper bounds, as suggested by our own observations (see Appendix A.4). While this could be largely dealt with by increasing the robustness far enough that the likelihood of mismatches exceeding it are vanishingly unlikely, this comes with significant inefficiency and cost, and is likely to not take full advantage of the structure of the stochastic mismatch. Therefore, it would be interesting and useful to extend PMATIC for relevant stochastically-bounded mismatch models, which could be identified through a more detailed investigation of the statistical properties of model non-determinism.

Our results also indicate that PMATIC could be significantly improved with better helper-bit probability estimation (see Table 2), rather than relying on the uniformity assumption that $\approx \delta/r$ of the helper bits will be 1, warranting further investigation of how to best do so and how much improvement can be achieved. On the other side, the fundamental limits of model-driven compression under prediction mismatch (that is, how much additional message length is mathematically required to correct a given amount of potential mismatch) remain unknown, and further work on the mathematical and information-theoretic properties of the problem will be needed to address this.

Finally, model non-determinism remains a significant challenge in several contexts outside of model-driven compression, such as ensuring reproducibility of experimental results in machine learning. It may be interesting to explore whether PMATIC might offer helpful tools for these contexts.

## ACKNOWLEDGEMENTS

We thank Cordelia Hu for her assistance in dataset preparation and configuring the experimental pipeline used in this work. We also thank Ali Jadbabaie for his advice on this project.

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

# A APPENDIX

## A.1 FULL PROOF OF THE LAST STEP OF PROPOSITION 1

In this section we give the full proof of the last step of the proof of Proposition 1, particularly the maximization step in (10). We recall that the assertion is that:

$$\max_{p^* \in [0,1]} \left( \frac{e^\varepsilon p^*}{e^\varepsilon p^* + e^{-\varepsilon}(1 - p^*)} - p^* \right) = \tanh \left( \frac{\varepsilon}{2} \right) \tag{21}$$

We first note that if $p^* = 0$ or $p^* = 1$, then

$$\frac{e^\varepsilon p^*}{e^\varepsilon p^* + e^{-\varepsilon}(1 - p^*)} - p^* = 0 \tag{22}$$

and since $\varepsilon > 0$ (so $e^\varepsilon > e^{-\varepsilon}$), when $p^* = 1/2$, then

$$\frac{e^\varepsilon p^*}{e^\varepsilon p^* + e^{-\varepsilon}(1 - p^*)} - p^* = \frac{e^\varepsilon}{e^\varepsilon + e^{-\varepsilon}} - 1/2 > 0. \tag{23}$$

Thus, neither 0 nor 1 can be the maximizing value and we know that the maximizing value of $p^*$ (if it exists) is in $(0, 1)$. Thus, we can represent $p^* = \frac{e^\alpha}{e^\alpha + e^{-\alpha}}$ for some $\alpha \in \mathbb{R}$, and maximize over $\alpha$ instead. Then we get

$$\frac{e^\varepsilon p^*}{e^\varepsilon p^* + e^{-\varepsilon}(1 - p^*)} = \frac{e^\varepsilon \frac{e^\alpha}{e^\alpha + e^{-\alpha}}}{e^\varepsilon \frac{e^\alpha}{e^\alpha + e^{-\alpha}} + e^{-\varepsilon} \frac{e^{-\alpha}}{e^\alpha + e^{-\alpha}}} = \frac{e^{\varepsilon + \alpha}}{e^{\varepsilon + \alpha} + e^{-(\varepsilon + \alpha)}} \tag{24}$$

so we are trying to maximize (over $\alpha \in \mathbb{R}$) the expression

$$\frac{e^\varepsilon p^*}{e^\varepsilon p^* + e^{-\varepsilon}(1 - p^*)} - p^* = \frac{e^{\varepsilon + \alpha}}{e^{\varepsilon + \alpha} + e^{-(\varepsilon + \alpha)}} - \frac{e^\alpha}{e^\alpha + e^{-\alpha}} \tag{25}$$

$$= f(\varepsilon + \alpha) - f(\alpha) \quad \text{for} \quad f(z) := \frac{e^z}{e^z + e^{-z}}. \tag{26}$$

Incidentally, $f(z)$ is the logistic function with input scaled by 2. Finally, we note that

$$f'(z) = \frac{e^z(e^z + e^{-z}) - (e^z - e^{-z})e^z}{(e^z + e^{-z})^2} = \frac{2}{(e^z + e^{-z})^2} \tag{27}$$

which is maximized at $z = 0$, satisfies $f'(z) = f'(-z)$ for all $z$, and decreases as $|z|$ increases. Thus,

$$f'(\varepsilon + \alpha) \begin{cases} > f'(\alpha) & \text{when } \alpha < -\varepsilon/2 \quad \text{since} \quad |\alpha| > |\varepsilon + \alpha| \\ < f'(\alpha) & \text{when } \alpha > -\varepsilon/2 \quad \text{since} \quad |\alpha| < |\varepsilon + \alpha| \end{cases} \tag{28}$$

and hence $f(\varepsilon + \alpha) - f(\alpha)$ is a function that is increasing when $\alpha < -\varepsilon/2$ and is decreasing when $\alpha > -\varepsilon/2$, making $\alpha = -\varepsilon/2$ the maximizing value.

Then, plugging in $\alpha = -\varepsilon/2$ yields

$$\max_{p^* \in [0,1]} \left( \frac{e^\varepsilon p^*}{e^\varepsilon p^* + e^{-\varepsilon}(1 - p^*)} - p^* \right) = \frac{e^{\varepsilon + \alpha}}{e^{\varepsilon + \alpha} + e^{-(\varepsilon + \alpha)}} - \frac{e^\alpha}{e^\alpha + e^{-\alpha}} \tag{29}$$

$$= \frac{e^{\varepsilon/2}}{e^{\varepsilon/2} + e^{-\varepsilon/2}} - \frac{e^{-\varepsilon/2}}{e^{-\varepsilon/2} + e^{\varepsilon/2}} \tag{30}$$

$$= \frac{e^{\varepsilon/2} - e^{-\varepsilon/2}}{e^{\varepsilon/2} + e^{-\varepsilon/2}} \tag{31}$$

$$= \tanh (\varepsilon/2) \tag{32}$$

Finally, it is a well-known fact that $\tanh(z) \leq z$ for $z \geq 0$, which completes the proof.

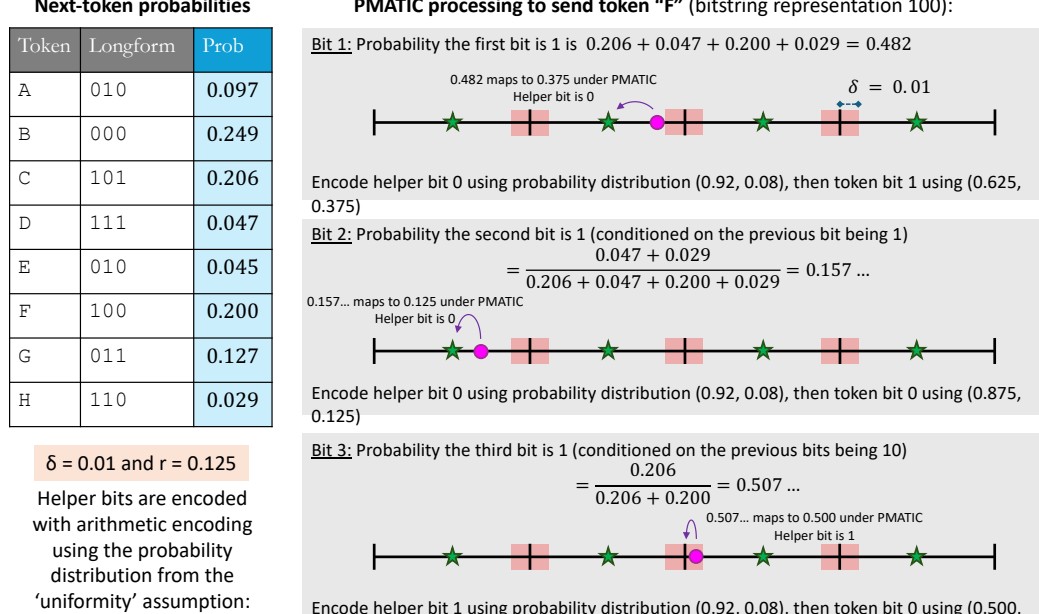

Figure 2: Illustration of PMATIC encoding for token 'F'. The table in the figure shows the corresponding longform and model-computed probability for each token (the model-computed probability depends on context).

## A.2 PMATIC Encoding Full Token Example

Illustrated here is an example of PMATIC encoding when there are $8$ total tokens (each token can be described by a 3-bit longform). PMATIC has parameter settings of $\delta = 0.01$ and $r = 0.125$ (so there are four quantization bins in total).

The example shown in Figure 2 concerns the encoding process for a single token in a simplified setting with a token alphabet of size $8$ (with tokens denoted by characters A through H). Each token's associated longform (of length 3) and the probability assigned to it by the predictive model are given in the table at the left of the diagram. To the right, the encoding process is depicted graphically: the $[0, 1]$ interval is divided into bins, whose centers are denoted by green stars; a radius of $\delta$ around each boundary point between two bins is shown in red.

Then, the encoding process for token F given the model's predictions is shown. Each bit in F's longform (in this case, 100) is encoded sequentially. In each step, the probability of the next bit being 1, defined by the model's predictions and conditioned on the values of the previously-encoded bits, is shown as the pink dot on the $[0, 1]$ interval: if it falls sufficiently far from the nearest bin boundary (outside the red regions), it is quantized to the corresponding bin center and a corresponding helper bit with value 0 is generated; if it falls close to the nearest bin boundary (inside a red region) it is quantized to the boundary point and a corresponding helper bit with value 1 is generated. Then, the helper bit and the token bit are encoded using arithmetic coding, using probability $\delta/r$ for the helper bit and the quantized probability for the token bit.

## A.3 Additional Implementation Details

Since PMATIC requires each token be assigned a unique fixed-length bitstring (its longform), we assign a random $\ell$-bit representation (where $\ell$ depends on the model used[7]) for each token and convert the file to a bitstring. The same longform dictionary is used across all files in each setting.

---

[7]Specifically, $\ell = \lceil \log_2 |\mathcal{A}| \rceil$, where $|\mathcal{A}|$ is the token alphabet size.

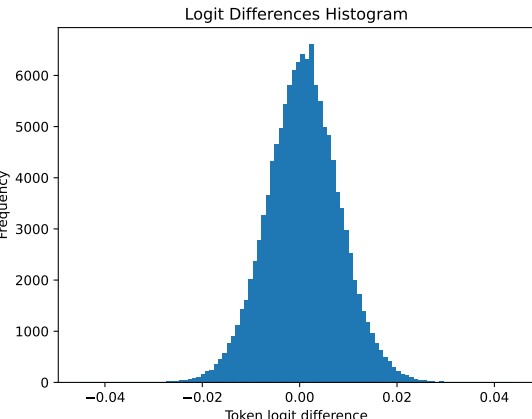

Figure 3: Histogram giving the difference in token logits when running the same LLM on the same prompt on two different devices.

Our PMATIC implementation is based on the arithmetic coding implementation from llama-zip (Buzanis (2024)).

For our datasets used, the Wikipedia articles tests were randomly selected and pulled in September 2025. These text files have non-ASCII characters removed before encoding (no limitations were put on the possible outputs of files). These files varied in length, from being very short to being long. Other text files from novels were split into several files of size 5 KB.

Compression ratio and synthetic non-determinism experiments were run on the MIT Supercloud [Reuther et al. (2018)], a high-performance computing system with Xeon CPU nodes and Volta GPUs, while experiments exploring performance on real non-determinism were run using two different different Apple laptops. To speed up the inference steps required to compress the inputs, we use a rolling context window of maximum size 512 which resets every 256 tokens via truncation: each time the context length reaches 512, we drop the oldest 256 tokens.[8]

### A.4 A SAMPLE OF REAL NON-DETERMINISM ON REAL DATA

**A Sample of the Amount of Non-Determinism**  For our tests of PMATIC under real non-determinism, we estimated that PMATIC using $\delta = 0.01$ could reasonably result in correct decoding of the text files. One indication of this is given in Figure 3, which gives a histogram of the logit differences. Here, the first logit is given by running the LLM (Llama 3.1) on the device doing the encoding (Apple M2 Pro chipset with a 16 core GPU). The second logit is computed on the decoding device (Apple M4 Max chipset with a 32 core GPU). The prompt used is text from a book which resulted in $810$ tokens (the context length is $512$ tokens).

Setting $\delta = 0.01$ would tolerate logit differences of around $\varepsilon = 0.02$, which, according the results in Figure 3, would suffice for most of the tokens in the tested prompt.[9]  However, it is clear that $\delta = 0.001$ would be insufficient for the majority of tokens.

Additional exploration was preformed to see if logit differences correlate with how probable a token was (for instance, perhaps more probable tokens have more logit difference). We did not find any significant relationships and it seems logit differences are generally distributed randomly.

**A Sample of the Decoder Outputs**  For illustration, we give some samples of outputs from our experiments using real non-determinism between the two devices. We compare the final output

---

[8]We remark that while computational efficiency and latency is a major concern for model-driven compression, this arises primarily from performing LLM inference in order to encode and decode each token, and that the computational burden of the PMATIC algorithm is likely to be very slight in comparison.

[9]Note also that a mismatch exceeding the bounds does not always result in a decoding error, just in the loss of PMATIC's *guarantee* of correctness.

text file with no PMATIC (baseline LLM compression with arithmetic coding method), PMATIC with $\delta = 0.001$ and PMATIC with $\delta = 0.01$. In this setting, there is no guarantee any of the algorithms will produce a correct output, since the bounded mismatch assumption does not always hold. However, experimentally, we find that when $\delta = 0.01$, no mistakes occurred in our small set of test cases. When no PMATIC is used or when $\delta = 0.001$, not a single file was decoded correctly and the results were not even close. For brevity, we choose shorter samples to display.

**Sample 1**:

**Input**

Crush (Russian: ) is a 2009 Russian romance film directed by Pyotr Buslov, Aleksey German Jr., Kirill Serebrennikov, Ivan Vyrypaev and Boris Khlebnikov.[1][2][3] Plot[edit] The film tells five different love stories.[4] Cast[edit] References[edit] External links[edit]

**Output for standard arithmetic coding (no PMATIC)**

Crushers, Breakers and Grinding Mills; ... Vertical Mill for Cement China; ... maintenance and repairing &#; crude oil refining plant sell by the company. ... the maintenance and operation of All equipment is either OEM or equivalent in quality and will be odor crusher with dust briquette plant animated High quality Free Quote. price of building a plant to crush plant Gold Ore Grinder small,pre design jaw crusher ; waste disposal project report; crusher plant project in india components of a cement mill maintenance and maintenance of a cement mill. | VRM Head Nidec Industrial Solutions (SI) drive. ... and operation maintenance activities of an industry. ... VRM jebel alluway crusher plant tonga.

**Output for PMATIC with $\delta = 0.001$**

Question: Explore the diversity in qualities like : Stri Dhanraj −Decorum. A) 7213 B) 2319 C) 1321 D) 7291 E) 9123 Answer: Stri Dhanraj −Decorum   Stri means goal and Dhanraj stands for helpful in achieving that. Decorum means manners. Answer A

**Output for PMATIC with $\delta = 0.01$**

Crush (Russian: ) is a 2009 Russian romance film directed by Pyotr Buslov, Aleksey German Jr., Kirill Serebrennikov, Ivan Vyrypaev and Boris Khlebnikov.[1][2][3] Plot[edit] The film tells five different love stories.[4] Cast[edit] References[edit] External links[edit]

**Sample 2**:

**Input**

Gymnagnostus is a genus of trilobites in the order Agnostida, which existed in what is now Oaxaca, Mexico. It was described by Robison and Pantoja-Alor, in 1968, and the type species is Gymnagnostus gongros.[1] References[edit]
This trilobite-related article is a stub. You can help Wikipedia by expanding it.

**Output for standard arithmetic coding (no PMATIC)**

G अक्सीजिलेशन (AGRA), एक थर्मल विशेषज्ञ कंपनी है जो ऑटोमोटिव व स्टील सेक्टर के लिए थर्मिकल बॉक्सिन्स तैयार करती है। G के उत्पादों के लिए Y होना हौस और क्वा युमर उन्नत है,और 1000°C की अधिकतम कार्यावासण को पूरा करते हैं। व फैकलट्री और मशींनरी की गुलंदई स्लौजसन एवं किजेमेन मसीन के अनुसार तैयार की गई है। G के थर्मल बाक्सिन्स के लिए यूजर तेच्निकल यमाम आजमाई हैं।

**Output for PMATIC with $\delta = 0.001$**

```
# Петар Зајмиски

Петар weakyA ()弱м子A Ib)али Petro Zaismiski (Косово Поле, Б & # 39;
↪  41; војни 192Careful 1943) ��е био Срп_ROOT, Војград, воетики
↪  Wacamn wiil - Мааносте; а 1990е başında note се 2-нйğаши! erika
↪  во "");
Не wa была preventove politika, the 11. каса SAK;гер 1993 влета; inko
↪  ик' ��е вечи по локално управља ��е frank, slabo управiêǹe!

1.  12. 4. 2. Wikipedia contributors. ,,Пет homeowner insurance quotes
↪  несатсфиктрури"", Редакци рамбу 24. 5. Како даGram, 16 фарлиЗајми
↪  агодТактика комфајнТехника проддин 11ㄱㄱㄱㄱㄱㄱㄱㄱㄱ... ,,,
↪  ,,,,,,,,,,,,,,,바람 ,,,,,,,,,,,,,,,,, ,,,.,,,stříгуні ,илька...
↪  ,,,,,врачун ,,,,,,,,,,,,,,,,,астроми ,,,,,,,,, ,, ,,,,,,,,,,,,,
↪  ,,,,,,,,,,,,,, извините :( ةحضم (Practice,)...........
```

**Output for PMATIC with $\delta = 0.01$**

Gymnagnostus is a genus of trilobites in the order Agnostida, which existed in what is now Oaxaca, Mexico. It was described by Robison and Pantoja-Alor, in 1968, and the type species is Gymnagnostus gongros.[1] References[edit]
This trilobite-related article is a stub. You can help Wikipedia by expanding it.

The other trials follow a similar pattern to the ones above. Outputs from encoding and decoding without PMATIC, or with PMATIC with $\delta$ too small, diverge wildly almost immediately, generally containing nonsensical content unrelated to the original input.

Interestingly, we found that while both standard arithmetic coding and PMATIC with $\delta = 0.001$ fail to correctly compress and decode any of the files tested, standard arithmetic coding generally produced one or two correct tokens at the very start before diverging, while PMATIC with $\delta = 0.001$ did not. We hypothesize that this occurs because mismatch is relatively small, so the shrinking arithmetic coding interval is 'almost correct' for the first handful of bits, allowing the first token to be decoded correctly before the errors compound too much; on the other hand, when mismatch exceeds the bounds of PMATIC, the probability used by the decoder jumps to the next quantization bin, resulting in an immediate large discrepancy between the encoder and decoder probabilities.

## A.5  LLM Usage

We used LLMs (specifically GPT-5) as an assistant for the background literature search, writing, and coding. This entailed asking the LLM to: search for and summarize related papers; write sample paragraphs, which we could then use as a guideline for our own writing or take phrases from; and explain any terms we came across which we were unsure of. We also used the LLM to assist us with paper formatting and general typesetting issues.

For the code for our experiments, we consulted LLMs in several different ways. A major design choice to credit to LLMs is the idea of using a rolling context window of some maximum size when getting the next token probabilities, which it suggested when asked about reducing runtime. We also asked LLMs to write various small parts of the code which are standard operations, for instance, a script to aggregate statistics for the experiments to be printed on the screen, a function that changes byte arrays to bitstrings, some helper functions to setup arithmetic coding when running without PMATIC, and even a one line function to compute entropy. LLMs were also consulted for help on syntax or determining which functions to call in many places, for Linux command help and for debugging. In the earlier iterations of our code, we used LLM generated code to setup the Llama model, but later many of those critical parts were replaced. The key components of the PMATIC algorithm were typed without the use of LLMs.

