# OpenReview forum: "Synchronizing Probabilities in Model-Driven Lossless Compression"
_ICLR.cc/2026/Conference — ICLR 2026 Poster_

### Official Review · Reviewer_YPjH · 2025-10-18

**Soundness:** 3
**Presentation:** 2
**Contribution:** 3
**Rating:** 4
**Confidence:** 4

**Summary:**

This paper tackles a practically critical but underexplored problem in model-driven lossless compression: the instability caused by prediction mismatch between encoder and decoder due to LLM non-determinism. The authors formalize this as the probability matching problem and propose PMATIC (Probability-Matched Interval Coding), a theoretically grounded and model-agnostic alternative to arithmetic coding that tolerates bounded prediction mismatch. Theoretical guarantees on decodability and compression efficiency are presented, and experiments on Wikipedia and Enwik8 datasets validate correctness and reasonable efficiency loss under synthetic noise.

**Strengths:**

- The work highlights a key yet overlooked obstacle in deploying LLM-based compression systems: non-deterministic inference, bridging the gap between theory and practical system reliability.

- The formalization of bounded prediction mismatch and the introduction of a matching interval coding mechanism are conceptually clean and mathematically sound. The paper also provides provable correctness and upper bounds on the additional code length, demonstrating an informed trade-off between robustness and compression efficiency.

- PMATIC is model-agnostic and can be seamlessly integrated as a drop-in replacement for arithmetic coding, showing potential broad applicability beyond text compression.

**Weaknesses:**

- Experiments are confined to synthetic perturbations on a single model (Llama-3.1) and small text corpora. This does not convincingly capture the stochastic, architecture- or library-dependent non-determinism that motivates the problem. I'd like to see more experiments.

- There already exist benchmarks for model-driven or LLM-assisted compression [1,2,3]. These could provide a valuable performance baseline.

- Runtime, helper-bit statistics, and computational overhead are not reported. It is unclear how PMATIC scales when applied to larger models or real-time compression.

- Although the paper derives an analytical bound on the extra bit cost, it does not verify whether the empirical losses follow this bound.

- Some related works are missing [4,5,6]

- In all, I think this paper addresses an important and practical problem, but the experimental section is insufficient. If the authors can provide comprehensive and convincing experimental results during the rebuttal period, I will raising my score.

[1] Valmeekam C S K, Narayanan K, Kalathil D, et al. Llmzip: Lossless text compression using large language models[J]. arXiv preprint arXiv:2306.04050, 2023.
[2] Mao Y, Pirk H, Xue C J. Lossless Compression of Large Language Model-Generated Text via Next-Token Prediction[J]. arXiv preprint arXiv:2505.06297, 2025.
[3] Mittu F, Bu Y, Gupta A, et al. Finezip: Pushing the limits of large language models for practical lossless text compression[J]. arXiv preprint arXiv:2409.17141, 2024.
[4] Mao Y, Li J, Cui Y, et al. Faster and stronger lossless compression with optimized autoregressive framework[C]//2023 60th ACM/IEEE Design Automation Conference (DAC). IEEE, 2023: 1-6.
[5] Mao Y, Cui Y, Kuo T W, et al. Accelerating general-purpose lossless compression via simple and scalable parameterization[C]//Proceedings of the 30th ACM International Conference on Multimedia. 2022: 3205-3213.
[6] Goyal M, Tatwawadi K, Chandak S, et al. DZip: Improved general-purpose loss less compression based on novel neural network modeling[C]//2021 data compression conference (DCC). IEEE, 2021: 153-162.

**Questions:**

- How large is the helper-bit overhead in practice, and how does it vary with δ?

- What is the computational impact, such as encoding/decoding latency compared to standard arithmetic coding?

- The paper mentions the potential extension to stochastically bounded mismatch. Could the authors elaborate on whether PMATIC can be adapted to probabilistic mismatch distributions observed in real inference pipelines?

---

> ### Author Response · Authors · 2025-12-03
> **Response to reviewer YPjH**
>
> We thank the reviewer for the detailed and constructive feedback and questions. We address their concerns and questions below.
>
> CONCERNS:
>
> 1. *Experiments are confined to synthetic perturbations on a single model (Llama-3.1) and small text corpora. This does not convincingly capture the stochastic, architecture- or library-dependent non-determinism that motivates the problem. I'd like to see more experiments.*
>
> We agree with the reviewer that an analysis of the size and distribution of inference mismatch is critical to understanding the problem and for analyzing the utility of PMATIC and how we might extend or improve it; however, such an analysis is beyond the scope of this work, which focuses on the theoretical framework and algorithm as its main contributions. See #1 of our general response for a more detailed discussion.
>
> 2. *There already exist benchmarks for model-driven or LLM-assisted compression [1,2,3]. These could provide a valuable performance baseline.*
>
> We agree that prior work on LLM-assisted compression provides valuable insight and context on what may be achievable using these techniques. However, these prior works use a variety of different models to drive compression and focus on the model inference step, making direct comparison inappropriate. See #3 of our general response for a more detailed discussion.
>
> 3. *Runtime, helper-bit statistics, and computational overhead are not reported. It is unclear how PMATIC scales when applied to larger models or real-time compression. Although the paper derives an analytical bound on the extra bit cost, it does not verify whether the empirical losses follow this bound.*
>
> We agree that latency is a major concern for LLM-driven compression algorithms. However, our implementation of PMATIC is not optimized for computational efficiency, and hence we believe that including latency figures for the current version would be uninformative. Furthermore, LLM-driven compression pipelines typically have their latency bottleneck at the model inference step; therefore, we expect that PMATIC will have a negligible impact on end-to-end latency. We will clarify this point in the revision. See point 2 of our main response for a more detailed discussion.
>
> Regarding helper-bit statistics, see question 5 below, and #4 in our gneeral response.
>
> Finally, to address the reviewer’s concerns about how PMATIC behaves when used with other models, we have run additional experiments with two other models (Mistral 7B v0.1 quantized and Qwen 2.5 Instruct 7B quantized), which confirm our results with Llama-3.1 8B quantized. See #1 of our general response for a more detailed discussion.
>
> 4. *Some related works are missing [4,5,6]*
>
> We thank the reviewer for drawing our attention to these works and will add them as citations.
>
> QUESTIONS:
>
> 5. *How large is the helper-bit overhead in practice, and how does it vary with δ?*
>
> We have looked at the helper-bit statistics and found that helper-bit entropy is, in practice (on text data), considerably smaller than the bound given by our theoretical assumption, which is designed to be “worst-case within reason” (assuming the next-bit probabilities don’t adversarially cluster around the boundaries between bins, but otherwise making no assumptions). This is to the advantage of PMATIC, and suggests a path for future work to significantly improve its effectiveness by tuning the position and number of bins to fit the domain. We will clarify this, add our results on helper-bit statistics, and include a discussion of the implications. See #4 of our general response for a more detailed discussion.
>
> 5. *What is the computational impact, such as encoding/decoding latency compared to standard arithmetic coding?*
>
> See our response to Concern 3 above, and #2 of our general response for more detail.
>
> 6. *The paper mentions the potential extension to stochastically bounded mismatch. Could the authors elaborate on whether PMATIC can be adapted to probabilistic mismatch distributions observed in real inference pipelines?*
>
> While PMATIC currently only guarantees correctness under deterministically bounded mismatch, extending it to cover cases of stochastically bounded mismatch is an open direction we are actively exploring. This includes approaches based on: (1) detecting and correcting errors quickly as they occur during decoding; (2) using stochastic concentration bounds to strongly bound cumulative mismatch with very high probability; (3) varying bin number and positions depending on the token distribution or domain characteristics. We will add a short discussion of this problem in the final version.

---

### Official Review · Reviewer_iuaJ · 2025-10-23

**Soundness:** 3
**Presentation:** 3
**Contribution:** 3
**Rating:** 8
**Confidence:** 3

**Summary:**

We theoretically formulated the problem of "non-determinism" that arises in lossless compression using large-scale language models (LLMs), and proposed a new compression coding method, PMATIC (Probability-Matched Interval Coding), to overcome this problem.
LLM outputs a probability distribution $P(x_t|x_{<t})$, but even for the same input, it may produce slightly different probabilities due to different GPUs, different libraries, or differences in parallel order. Conventional arithmetic coding breaks down if the probability distributions of the encoder and decoder do not match perfectly. However, if the probability mismatch is sufficiently small (bounded mismatch), accurate decoding is possible if both agree on an "intermediate common distribution", which is the Probability Matching problem.
PMATIC is a method that extends existing arithmetic coding with probability quantization, ensuring consistent coding even if the encoder and decoder make slightly different probability predictions.
For PMATIC, this paper theoretically evaluates its correctness and compression loss.

**Strengths:**

Originality: This paper is the first to mathematically formalize the problem of "probabilistic model-driven compression" $\times$ "LLM nondeterminism".
Quality: Not only is there a mathematical discussion of performance analysis, but the usefulness of the proposed method is also verified through numerical experiments, making the paper of high quality.
Clarity: The problem to be addressed is clearly stated, the algorithm is given in detail, and the argument is clear enough.
Significance:  This paper is one of the first to formulate the problems that are encountered when actually applying LLM as lossless compression, and is expected to make an important contribution to related research.

**Weaknesses:**

As mentioned in 6. Future work, if LLM is actually used for lossless compression, the size of the models required to implement the compressor and decoder will be a major problem. This issue has not been discussed in this paper.

**Questions:**

There has been discussion about static performance, such as the achievable compression performance. However, how much better is it compared to existing methods in terms of the amount of computation required for compression and decoding?

---

> ### Author Response · Authors · 2025-12-03
> **Response to reviewer iuaJ**
>
> We thank the reviewer for their detailed and constructive feedback.
>
> CONCERNS
>
> 1. *As mentioned in 6. Future work, if LLM is actually used for lossless compression, the size of the models required to implement the compressor and decoder will be a major problem. This issue has not been discussed in this paper.*
>
> We agree with the reviewer that model size is a major challenge to fielding practical model-driven lossless compressors. However, our work aims to address a different challenge (model non-determinism) which is orthogonal to reducing the memory footprint of the model driving the compression. Development of lightweight (and computationally efficient) models is a highly active area of research across ML and it is our belief that advances in that area can then be combined with mismatch-robust coding algorithms like PMATIC to produce practically-viable model-driven compression algorithms across many different types of data.
>
> QUESTIONS
>
> 2. *There has been discussion about static performance, such as the achievable compression performance. However, how much better is it compared to existing methods in terms of the amount of computation required for compression and decoding?*
>
> As the reviewer alludes to, it is well-known that many traditional codecs are significantly faster than model-driven algorithms, due to the need (in model-driven compression) to run a forward pass at every token. However, PMATIC does not affect the underlying model and is instead a coding algorithm based on arithmetic coding which is designed to work with any predictive model and to be able to handle (bounded) inference non-determinism. Thus, latency improvements for model-driven compression are primarily dependent on latency improvements for predictive models in general, which is a huge and extremely active area of research across ML and is outside the scope of this work. See #2 of our general response for a more detailed discussion.

---

### Official Review · Reviewer_nkTr · 2025-10-31

**Soundness:** 3
**Presentation:** 1
**Contribution:** 4
**Rating:** 6
**Confidence:** 4

**Summary:**

The authors propose an algorithm to quantize / bin the probability distributions to account for possible deviations of the predicted CDFs between the encoder and decoder. The motivation is due to the non-determinism of LLMs, which can result in small differences in floating point numbers even when run on the same hardware.

**Strengths:**

- The method is reasonably simple to apply to existing LLMs, without needing to retrain.
- The authors provide some guarantees on the added bit length due to binning.
- This is a very important, and realistic problem, that needs to be solved to have next-gen AI codecs, and I appreciate the authors pushing on real world problems.

**Weaknesses:**

- The experiments are significantly lacking in breadth. If the method is general, the authors could provide further experiments with different data modalities.

- The following is hard, but would significantly improve the paper: can the authors estimate what are typical deviations present in relevant scenarios where AI codecs could be applied? For example, take any open source model, and apply the encoder and decoder using 1) a different version of CUDA, 2) different models, and other variables that might vary in practice. This would significantly improve the contribution and ground the paper in real world applications.

- Adding figures explaining the binning procedure would significantly improve the exposition of the algorithm.

**Questions:**

- Do the authors have a good sense of how large the mismatches are in practice (see box above)?
- Can the authors provide more experiments across data modalities, and possibly varying other variables that could impact the degree of mismatch (see box above)?
- Can the authors add figures to explain the binning better?

---

> ### Author Response · Authors · 2025-12-03
> **Response to reviewer nkTr**
>
> We thank the reviewer for their detailed and constructive feedback.
>
> CONCERNS
>
> 1. *The experiments are significantly lacking in breadth. If the method is general, the authors could provide further experiments with different data modalities.*
>
> We agree that further experiments showing the performance of mismatch-robust model-driven compression on different modalities would be extremely valuable, and view it as a key component of future work on this topic. However, it is outside the scope of this work, which focuses on the problem statement, theoretical framework, algorithm, and proof of concept experiments as its main contribution. To expand the breadth of the experiments, we ran additional tests using other text models and data (including texts in languages other than English). See #1 of our general response for a more detailed discussion.
>
> 2. *The following is hard, but would significantly improve the paper: can the authors estimate what are typical deviations present in relevant scenarios where AI codecs could be applied? For example, take any open source model, and apply the encoder and decoder using 1) a different version of CUDA, 2) different models, and other variables that might vary in practice. This would significantly improve the contribution and ground the paper in real world applications.*
>
> We agree with the reviewer that an analysis of the size and distribution of inference mismatch is critical to understanding the problem and for analyzing the utility of PMATIC and how we might extend or improve it; however, as the reviewer notes, such an analysis could be a research project on its own and is beyond the scope of this work, which focuses on the theoretical framework and algorithm as its main contributions. See #1 of our main response for a more detailed discussion.
>
> 3. *Adding figures explaining the binning procedure would significantly improve the exposition of the algorithm.*
>
> We thank the reviewers for this suggestion, and have added a figure (Figure 1) explaining the binning procedure to the revised version, as well as an expanded figure (Figure 4) giving a full example of a token encoding with PMATIC in the appendix.
>
> QUESTIONS
>
> All questions refer to the concerns above.

---

### Official Review · Reviewer_ADbL · 2025-11-08

**Soundness:** 4
**Presentation:** 4
**Contribution:** 3
**Rating:** 8
**Confidence:** 4

**Summary:**

The paper proposes a method to address the problem of practical usage of LLMs in data compression. A novel algorithm is presented for compression with uncertainty in probability predictions based on arithmetic coding. Furthermore mathematical analysis is presented showing compression bounds under a given uncertainty. Experiments denote successful demonstration with synthetically generated noise.

**Strengths:**

1. This is a very relevant problem, I believe this would allow the community to actually make practical compressors with the proposed algorithm.
2. Article is original and a novel algorithm is proposed. Paper is quite easy to read.

**Weaknesses:**

1. The only weakness I would say is some analysis on what delta's would we expect by changing hardware or going from GPUs to CPUs. Also there is relevant research addressing uncertainty in LLM prediction which should be added to related work (https://thinkingmachines.ai/blog/defeating-nondeterminism-in-llm-inference/).
2. Please add comparison with some more compressors, especially CMIX. Also include latency because that is where traditional compressors win by a huge margin.

**Questions:**

Please see weaknesses.

---

> ### Author Response · Authors · 2025-12-03
> **Response to reviewer ADbL**
>
> We thank the reviewer for the detailed and constructive feedback and questions. We address their concerns below.
>
> 1. *The only weakness I would say is some analysis on what delta's would we expect by changing hardware or going from GPUs to CPUs. Also there is relevant research addressing uncertainty in LLM prediction which should be added to related work (https://thinkingmachines.ai/blog/defeating-nondeterminism-in-llm-inference/).*
>
> We agree with the reviewer that an analysis of the size and distribution of inference mismatch is critical to understanding the problem and for analyzing the utility of PMATIC and how we might extend or improve it; however, such an analysis is beyond the scope of this work, which focuses on the theoretical framework and algorithm as its main contributions. See #1 of our general response for a more detailed discussion.
>
> Prior work on addressing LLM inference non-determinism mainly focuses on how LLMs can be modified to reduce or eliminate non-determinism, while our approach is to address non-determinism at the coding stage through a mismatch-robust codec. These approaches can be complementary. We will include an expanded discussion on these prior works and this distinction in the revised version. See #3 of our general response for a more detailed discussion.
>
> 2. *Please add comparison with some more compressors, especially CMIX. Also include latency because that is where traditional compressors win by a huge margin.*
>
> We have added a number of other modern compressors, including CMIX, as further baselines for comparison; the results are consistent with prior literature indicating that LLM-driven compressors can achieve better compression ratios than traditional compressors, and show that LLM-driven compressors outperform traditional compressors even with the robustness-guarantee overhead from PMATIC. See #3 of our general response for a more detailed discussion.
>
> Regarding latency, we agree that this is a major concern for LLM-driven compression algorithms. However, our implementation of PMATIC is not optimized for computational efficiency, and hence we believe that including latency figures for the current version would be uninformative. Furthermore, LLM-driven compression pipelines typically have their latency bottleneck at the model inference step; therefore, we expect that PMATIC will have a negligible impact on end-to-end latency. We will clarify this point in the revision. See #2 of our general response for a more detailed discussion.

---

### Author Response · Authors · 2025-12-03
**Revision updated with new experiments and figures**

We thank all reviewers for their thoughtful and constructive feedback. We have begun incorporating these into a revised manuscript, which has now been uploaded. This revision remains a work-in-progress, and not all planned additions have been included; however, key additions such as the results tables from expanded experiments and new explanatory figures have been added.

---

### Author Response · Authors · 2025-12-03
**General response to reviewer comments #1**

1. *Several reviewers express an interest in seeing an expanded suite of experiments, including experiments utilizing different models to drive compression, different modalities, and experiments studying or using naturally-occurring mismatches.*

We appreciate the reviewers’ comments regarding the breadth of the experimental evaluation, and agree that fully confirming the utility of PMATIC (or subsequent mismatch-robust coding algorithms) will require more extensive validation with a variety of modalities and models, and with naturally-occurring non-determinism. Our experiments were intentionally designed to validate the theoretical results and our PMATIC algorithm, which are our main contribution, under controlled conditions where the mismatch can be precisely calibrated, and to directly compare PMATIC against standard arithmetic coding, which is the appropriate no-robustness baseline.

In response to the reviewers, we have run additional experiments using other models (Mistral 7B v0.1 (3-bit quantized, small) and Qwen 2.5 Instruct 7B (3-bit quantized, medium)) and additional text samples such as novels (included ones in French and Chinese). These results align with the findings reported for PMATIC on Llama-3.1-8B-quantized: LLM-driven compression achieves compression ratios ranging from ≈ 6% (Llama-3.1 on Emma by Jane Austen) to ≈ 13% (Qwen-2.5 Instruct 7B on Dream of the Red Chamber by Cao Xueqin), with PMATIC adding an additional overhead of 0.5%-1.8% (of the un-compressed length) to achieve robustness of δ=10^{-5}, overhead of 4.5%-7% to achieve robustness of δ=0.001, and overhead of 13.5%-20% to achieve robustness of δ=0.01. Furthermore, LLM-driven compression (for all LLMs) achieves better compression ratios than all traditional codecs across our dataset, even with the maximum additional robustness overhead from PMATIC.

For the full table including the results of all new experiments, see Figure 2 of our revised manuscript.

For real-pipeline nondeterminism, we fully agree that characterizing naturally occurring mismatch is important. Performing this evaluation rigorously requires a diverse suite of models, hardware configurations, and inference stacks (e.g., CUDA/cuBLAS/cuDNN versions, GPU vs CPU, parallelization strategies), as well as substantial measurement infrastructure. Moreover, as the reviewers point out, such scenarios generally lead to stochastic mismatch rather than the hard deterministic bound considered in this paper. Extending PMATIC to this setting requires additional theory, and we are actively developing such extensions in follow-up work, including approaches based on error detection and correction, concentration bounds for giving high-probability bounds on mismatch, and more sophisticated quantization binning based on the distribution of next-token probabilities.

We also agree that broader multimodal evaluation (e.g., vision models) would be valuable, but it is beyond what can be added during the revision; we view this as a natural direction for future work.

Overall, our goal in this work was to introduce a theoretically grounded framework for robust model-driven compression and to validate it in a clean, controlled setting. We view comprehensive empirical characterization of real-world nondeterminism, and adapting PMATIC to stochastic mismatch regimes, as exciting and substantial future directions. We will add the results from the additional experiments and an extended discussion of more realistic mismatch models (and how PMATIC might be extended to cover them) in the revised version.

---

### Author Response · Authors · 2025-12-03
**General response to reviewer comments #2**

2. *Reviewers asked for measurements of the computational cost of PMATIC compared to standard arithmetic coding. They note that PMATIC adds quantization and helper-bit handling, which may introduce encoding/decoding latency. Reviewers ADbL and iuaJ, in particular, highlight that traditional compressors are often orders of magnitude faster than LLM-driven compressors, and thus runtime may be an important practical consideration. They request quantitative runtime comparisons or discussion of computational impact, especially for real-time or large-scale usage.*

We agree that latency is an essential consideration for practical model-driven compression, and we appreciate the reviewers highlighting this point. Our implementation was designed as a research prototype to validate correctness and measure compression performance. It was not optimized for speed, and raw timing numbers from this unoptimized prototype would not meaningfully reflect the practical runtime of PMATIC in a production setting.

We emphasize, however, that the dominant contributor to latency in model-driven compression is the model inference step: each token requires a full forward pass at both the encoder and decoder. As such, PMATIC (which does not affect the inference step in the pipeline) is not expected to materially change end-to-end latency. Improving runtime for model-driven compression is thus fundamentally a problem of improving language-model inference efficiency, through smaller or specialized models, architectural or caching optimizations, or hardware acceleration; this problem is orthogonal to the problem PMATIC solves, and constitutes a major area of ongoing research across machine learning. We believe that the ongoing development of lightweight, efficient models will help address this problem and can then be combined with mismatch-robust coding algorithms like PMATIC to produce model-driven compression algorithms which are viable in practice.

Since the question about latency is primarily concerned about the latency of LLM-driven compression, we note that improving the latency of LLM-driven compression, and comparisons with traditional codecs, was studied by Mittu et al. (2024, FineZip) (as cited in the introduction of our original submitted draft).

We will clarify this point more explicitly in the manuscript, and provide the FineZip citation for readers interested to learn more. While we fully agree that computational efficiency is crucial for future deployment, optimizing inference speed or developing real-time systems lies outside the scope of this paper, which focuses on developing a framework for mismatch-robust compression, and in particular on the correctness and theoretical compression efficiency of PMATIC.

---

### Author Response · Authors · 2025-12-03
**General Response to reviewer comments #3**

3. *Several reviewers request comparisons to additional baselines. Reviewer ADbL explicitly asks for CMIX, and Reviewer YPjH notes that existing model-driven compression benchmarks (LLMZip, FineZip, etc.) should be included. Some reviewers frame this as necessary to contextualize PMATIC’s practical performance, while others imply that such baselines are important given that traditional compressors often achieve strong performance with excellent latency. Reviewer ADbL also points to prior research studying non-determinism in machine learning models.*

We appreciate the reviewers’ suggestions regarding additional baselines, including CMIX and recent model-driven compressors such as LLMZip and FineZip, and suggestions of relevant prior literature.

To improve our traditional-codec baseline set, we ran additional tests using CMIX and other modern codecs (such as brotli, bzip2 and others). CMIX performed the best in general (though it was slightly beaten on wikipedia data by Brotli) and significantly outperformed GZip; however LLM-driven compression (for all LLMs, on all data tested) still achieved better compression ratios than all the baseline traditional codecs we evaluated, even with the highest robustness setting of PMATIC that we tested. These results are consistent with prior literature showing that LLM-driven probability models outperform classical codecs when the predictive model is sufficiently strong, and have been added to the current revised draft.

Regarding recent work on model-driven compressors (such as LLMZip and FineZip): LLMZip (and similarly llamazip which we also cite in our work) shows that compression with LLM is possible and gives experimental results on what is achievable. FineZip is a follow-up work which improves upon the runtime of LLMZip. Both are highly relevant works and both were cited in the original manuscript in the introduction (LLMZip is the work of Valmeekam et al. (2023) and FineZip is the work of Mittu et al. (2024))). However, neither of these works considers the problem of non-determinism for LLM-driven compression, which is the key objective our work. For this reason, the natural and most informative baseline for PMATIC is standard arithmetic coding using the same underlying model, since this isolates the effect of mismatch-robust interval quantization from the effect of changing the model itself.

Furthermore, direct comparison to prior benchmarks is not straightforward, because they use different LLM architectures (e.g., Llama-1) and different preprocessing and compression pipelines. These differences would obscure the specific contribution evaluated in our work: how much robustness to prediction mismatch costs when the model is held fixed. We will clarify this distinction in the revised version.

Finally, prior work on characterizing and mitigating non-determinism in machine learning models typically focuses on modifying inference procedures or model implementations to reduce nondeterminism directly, whereas our work asks whether the effects of nondeterminism can instead be addressed at the coding level through mismatch-robust coding algorithms. These perspectives are complementary: methods that reduce inherent nondeterminism could, in principle, be combined with PMATIC to allow tighter mismatch bounds. We will incorporate a brief discussion of this connection and add the appropriate citations (including {He, Horace and Thinking Machines Lab, "Defeating Nondeterminism in LLM Inference", Thinking Machines Lab: Connectionism, Sep 2025.}, as suggested by Reviewer ADbL; {Yuan, Jiayi, et al. "Understanding and Mitigating Numerical Sources of Nondeterminism in LLM Inference." arXiv preprint arXiv:2506.09501 (2025).}; and {Yuan, Jiayi, et al. "Give Me FP32 or Give Me Death? Challenges and Solutions for Reproducible Reasoning." arXiv preprint arXiv:2506.09501 (2025).}).

Overall, we agree that broader comparisons are useful for contextualizing the general space of compression methods, but for analyzing the specific phenomenon studied here (robust coding under model mismatch) the most appropriate baseline is standard arithmetic coding under the same LLM predictions.

---

### Author Response · Authors · 2025-12-03
**General response to reviewer comments #4**

4. *Reviewers ask how large the helper-bit overhead is in practice, how it varies with the mismatch bound δ, and whether empirical losses match the theoretical upper bound. They express interest in understanding whether the theoretical bound is tight, loose, or representative in practical settings. Some reviewers also request more detailed reporting of helper-bit statistics.*

We thank the reviewers for highlighting the question of helper-bit overhead.

In this work, we make a theoretical assumption of uniformity within each quantization bin, which implies that the helper bit is 1 with probability δ/r and leads to a theoretical overhead of (δ/r)*log_2(r/δ) bits per helper bit. This assumption is based on the idea that when designing the compression algorithm, we have no knowledge of the next-token distribution that will arise in the chosen domain, and in some sense represents a “worst case within reason” assumption (we assume that the next-bit probabilities do not adversarially cluster around our bin boundary points but otherwise make no assumptions about their distribution).

In additional experiments, we found that this theoretical assumption is very pessimistic in practice (within the domain of text compression), and that helper bit entropy can be an order of magnitude smaller than the theoretical (δ/r)*log_2(r/δ) bits per helper bit. This is to PMATIC’s advantage, and suggests that PMATIC could, in principle, be made significantly more efficient by tuning r (and the probability used for encoding helper bits via arithmetic coding) to better match the next-bit distribution of the domain. However, doing so with confidence requires a more extensive study of the model and the intended domain, as well as theoretical extensions; we view this as a promising avenue for future work.

This discrepancy is likely due to a prominent nonuniformity in next-bit probabilities. In many cases, the next bit could be nearly determined (e.g., when earlier bits in the token have narrowed the plausible token down). Probabilities near 0 or 1 do not trigger helper-bit = 1 events, since they don’t fall near the boundary between two bins, reducing entropy.

It is not yet clear how broadly this phenomenon generalizes to other modalities, and we will note this limitation.
We will add these empirical findings and a discussion of their implications to the revised manuscript. Currently in the revised version, we included helper bit statistics in Figure 3.

---

### Meta-Review · Area_Chair_3i1N · 2026-01-09

**Summary:**

The paper introduces Probability Matching Interval Coding (PMATIC), a model-agnostic algorithm for reliable decoding in model-driven lossless text compression under inference non-determinism. The motivation is that large language models can serve as strong compressors by predicting next tokens accurately, but because the encoder and decoder may run on mismatched hardware or otherwise experience non-deterministic inference, their predicted probabilities can diverge. In that setting, "classic" lossless source coding methods such as arithmetic coding can fail due to the next-token prediction mismatch between the encoder and the decoder.

All reviewers found the problem setting interesting and the proposed algorithm (PMATIC) novel. The primary concerns raised by the reviewers were largely about experimental breadth and benchmarking, including limited comparisons to other compressors and limited evaluation results, along with some requests for missing related literature and clearer presentation.

After reading the paper and the reviews, I agree with several reviewers that the experimental results could be more comprehensive. However, this does not detract from the novelty of the setting and the strength of the core theoretical contribution. I encourage the authors to add additional experiments suggested by the reviewers (when feasible) to the camera-ready version -- this could increase the impact of the paper.

**Reviewer Concerns:**

Several concerns around motivation, clarity, and missing related work were addressed in the rebuttal, and the authors either incorporated changes or committed to adding them in the revised manuscript.

Reviewer ADbL was positive on the contribution and raised a key concern about missing comparisons, especially against CMIX. This was addressed by the authors by adding the requested comparison.

Reviewer nkTr also had a positive view but echoed a broader theme across reviews: the experiments lacked breadth, and additional benchmarking would strengthen the empirical case for PMATIC. The authors recognized their current experimental evaluation as more of a "proof-of-concept" of their theoretical contribution, and positioned broader comparisons as future work.

Reviewer iuaJ was favorable overall and raised a relatively minor practicality issue: the compressor and decoder rely on large models. This is a limitation shared by **any** LLM-based compressors and should not be held uniquely against this work.

The most critical feedback came from YPjH, who reiterated the limited experiments and lack of benchmarks. In my opinion, the point-by-point rebuttal addressed most of these concerns, though some experimental limitations remain only partially addressed (as mentioned above).

**Reviewer Scores:**

For iuaJ, ADbL, and nkTr, I expect their score would remain essentially unchanged or slightly increase, as their feedback was already very favorable and focused mainly on model size, lack of experiments (some of which was added to the paper), and general limitation of LLM-based compression.

For YPjH, who was the most negative reviewer, I believe their score would likely increase, or at minimum not decrease, because multiple concerns they raised about benchmarking and empirical support were directly addressed in the rebuttal.

---

### Decision · Program_Chairs · 2026-01-26

Accept (Poster)